# Addressed Combined Fiber-Optic Sensors as Key Element of Multisensor Greenhouse Gas Monitoring Systems

**DOI:** 10.3390/s22134827

**Published:** 2022-06-26

**Authors:** Oleg Morozov, Yulia Tunakova, Safaa M. R. H. Hussein, Artur Shagidullin, Timur Agliullin, Artem Kuznetsov, Bulat Valeev, Konstantin Lipatnikov, Vladimir Anfinogentov, Airat Sakhabutdinov

**Affiliations:** 1Department of Radiophotonics and Microwave Technologies, Kazan National Research Technical University Named after A.N. Tupolev-KAI, 10 K. Marx St., Kazan 420111, Russia; taagliullin@mail.ru (T.A.); aakuznetsov@kai.ru (A.K.); kje.student@mail.ru (B.V.); klipatnikov87@mail.ru (K.L.); v.anfinogentov@yandex.ru (V.A.); azhsakhabutdinov@kai.ru (A.S.); 2Department of General Chemistry and Ecology, Kazan National Research Technical University Named after A.N. Tupolev-KAI, 10 K. Marx St., Kazan 420111, Russia; juliaprof@mail.ru; 3Department of Physics, College of Education for Pure Sciences, University of Kerbala, Freiha St., Kerbala 56001, Iraq; safaa.mohammed@uokerbala.edu.iq; 4Research Institute for Problems of Ecology and Mineral Wealth Use of Tatarstan Academy of Sciences, 28 Daurskaya St., Kazan 420087, Russia; artur.shagidullin@tatar.ru

**Keywords:** environmental monitoring, greenhouse gases, multisensor system, combined fiber-optic sensors, fiber Bragg grating, addressed fiber Bragg structure, Fabry–Perot resonator, Karhunen–Loève transforms

## Abstract

The design and usage of the addressed combined fiber-optic sensors (ACFOSs) and the multisensory control systems of the greenhouse gas concentration on their basis are investigated herein. The main development trend of the combined fiber-optic sensors (CFOSs), which consists of the fiber Bragg grating (FBG) and the Fabry–Perot resonator (FPR), which are successively formed at the optical fiber end, is highlighted. The use of the addressed fiber Bragg structures (AFBSs) instead of the FBG in the CFOSs not only leads to the significant cheapening of the sensor system due to microwave photonics interrogating methods, but also increasing its metrological characteristics. The structural scheme of the multisensory gas concentration monitoring system is suggested. The suggested scheme allows detecting four types of greenhouse gases (CO_2_, NO_2_, CH_4_ and O_x_) depending on the material and thickness of the polymer film, which is the FPR sensitive element. The usage of the Karhunen–Loève transform (KLT), which allows separating each component contribution to the reflected spectrum according to its efficiency, is proposed. In the future, this allows determining the gas concentration at the AFBS address frequencies. The estimations show that the ACFOS design in the multisensory system allows measuring the environment temperature in the range of −60…+300 °C with an accuracy of 0.1–0.01 °C, and the gas concentration in the range of 10…90% with an accuracy of 0.1–0.5%.

## 1. Introduction

In environmental monitoring, fiber-optic sensors (FOSs) are of global importance. The growing interest in FOSs is due to their advantages in comparison with the electronic sensors. The FOSs have numerous advantages such as a small size, a low weight, a high speed of response to the changing gas concentration, indifference to the electric and magnetic noises, a remote sensing ability and a resistance to the harsh environmental conditions [1]. Originally, FOSs were developed as the point sensors. A point sensor is less effective compared to a distributed or quasi-distributed sensor array in the amount of information, since it performs data collection from the single control point. The different technologies were developed to implement multipoint and distributed sensing for this reason [2]. The fibers with etched cladding in the FBG area [3], the plastic optical fibers [4], the Fresnel interferometers with various sensitive coatings were proposed as the sensing elements. The multipoint sensors can also be implemented using the optical fiber sounding methods. These methods have wide functional capabilities, and most of them show a linear response. However, a number of them are very unstable (due to the interferometric nature of the measurements), and some of them require expensive peripheral equipment while others have low reliability in the fiber breaking case (due to the serial connection) [5].

Using the multimode interference effect (MMI) [6,7] is a promising method of FOS creation. The FOSs based on the MMI principle are convenient because their architecture is simple, and they are easy to manufacture and compact. In addition, the spectral response of this class of sensors works like a narrow bandpass filter. In its simplest form, the FOS based on MMI includes a section of the multimode fiber (MMF) placed between two single-mode fibers (SMFs). These are commonly referred to as “SMF–MMF–SMF” structures [8,9]. Even this simple design allows measuring various physical and chemical parameters such as temperature, humidity, vibration, salinity and pH [10]. The narrowband spectral response of FOSs based on MMI can be easily tuned by a wavelength, adjusting its optical and geometrical parameters. The FOSs based on MMI can be easily used to design a fiber-optic multisensory system by wavelength-division multiplexing. Although these sensor systems show high measurement sensitivity, the multisensory system of this class requires additional photonic devices to integrate the sensors with the tunable lasers to interrogate them. The additional complexity is in realization of the high reflectivity fiber ends, which is quite difficult to obtain in the infrared spectrum range.

In choosing the direction of sensor development for environmental monitoring, the focus is on the fiber-optic Fabry–Perot end resonators (FPRs) [11]. The FPR is the classic sensitive element of pressure [12] and gas or liquid [13,14,15] concentration sensors. An open FPR for gas concentration control is made by splicing capillary tubes or photonic crystal fibers with the SMF [16]. The open resonator design allows gas flowing freely into the resonator cuvette, which changes the air refractive index between the mirrors. Despite this, the open FPR does not allow measuring the concentration of numerous gases, including greenhouse gases because the change in its concentration is too small to noticeably change the air refractive index between the FPR mirrors. Thus, the key to designing the greenhouse gas sensitive element of the FPR is finding or creating the material, the refractive index of which is noticeably sensitive to the gas concentration variation [11].

The transparent materials that are sensitive to carbon dioxide (CO_2_), for example, can be used as the sensitive film of the FPR resonator, where carbon dioxide interacts with the film material and changes its refractive index [17]. However, in practice, the sensors using a functional material are known only in fiber-optic refractometers based on the light attenuation effect [18]. The light attenuation coefficient depends on the refractive index of the sensitive film material, which is defined by the interaction of the film material with the gas. Thus, the sensor design as the end FPR is characterized by simplicity, the use of inexpensive absorbing materials and durability. Their design is a relevant task due to the simplicity of production and use.

Combined fiber-optic sensors (CFOSs) based on the combination of FBG and FPR for simultaneous gas, temperature and pressure measurement have recently undergone considerable development [19]. As estimations show, a promising sensitive structure for environmental monitoring should consist of FPR in the form of a thin film at the end face of the optical fiber with FBG near it. The film refractive index is reversibly changed depending on the gas concentration. The interference pattern of the thin-film FPR is sensitive to all environmental parameters, particularly to the changes in temperature, pressure and gas concentration. The FBG spectrum mainly depends on the temperature, and it has a weak dependence on other external parameters changing. Thus, CFOSs allow one to simultaneously measure the temperature and gas concentration. The problem is that environmental pressure and humidity are usually left out of the measurement [20]. A united CFOS sensor architecture not only serves as an effective method for the single-point measurement of the gas concentration and temperature, but also has great potential for sensors multiplexing. However, the cost of the system increases significantly due to the switching to a wide spectral range, which requires the usage of an expensive optoelectronic interrogator to separate the spectral responses from FBG and FPR [21].

In order to eliminate the aforementioned problems while preserving the functional advantages, the current work proposes the usage of the addressed fiber Bragg structures (AFBSs) instead of conventional FBGs. An AFBS is a fiber Bragg structure whose optical response includes two narrowband optical frequencies, while their difference is constant and lies in the microwave frequency range [22,23,24]. The difference of two optical frequencies is named “address frequency”, and it must be unique for each AFBS in the sensor system. A characteristic feature of AFBSs is the invariance of the address frequency under deformation and temperature influence. This allows using the AFBS address frequency for multiplexing in the sensor array [22]. AFBSs perform the triple function in the fiber-optic sensor systems: a sensor; a two-frequency light generator; and a multiplexer. The central wavelength of AFBSs can be detected without scanning its spectral range. It is the AFBS key feature. The AFBS interrogation scheme is much simpler compared to a classical optoelectronic interrogation scheme. It consists of a broadband optical light source (e.g., a super luminescent diode), an optical filter with a predefined frequency response with an inclined profile and a photodetector. The AFBS interrogation principle allows combining several AFBSs with the identical central wavelengths and the different address frequencies into a unified measurement system [22,25].

The goal of this work was the task statement of designing the addressed combined fiber-optic sensors (ACFOSs) as the key element of the multisensory measurement system. ACFOSs can be simultaneously used as the temperature and gas concentration sensors. ACFOSs are based on AFBSs (instead of FBG) combined with FPR, with improved metrological characteristics and the possibility of their multiplexing and interrogation in the fiber-optic multisensory systems for greenhouse gas monitoring.

To achieve this goal, the following tasks were formulated:—The design of ACFOSs, based on AFBSs and FPR, with the analysis of their manufacturing technology (first section);—The analysis of the principles of the separate analysis of the AFBSs and FPR responses by temperature and gas concentration based on the sensitivity matrix (second section);—The analysis of the ACFOS multiplexing principles based on AFBSs to address the properties and additional spectral analysis of the AFBSs and FPR responses to eliminate their cross-distortions using Karhunen–Loève transform (third section);—The design of the structural scheme and the interrogation principle for the multisensory system for the environmental monitoring of the greenhouse gases (the fourth section).

Formulating tasks for further research themes is the focus of our conclusion.

## 2. ACFOS Model

The structural diagram of ACFOSs is shown in Figure 1. An ACFOS can be represented as a layered structure consisting of three different layers for broadband laser light propagation: the fiber core; AFBSs, consisting of the two homogeneous FBGs with close but not equal central frequencies; and FPR. The thin film of the transparent organic polymer material, with a thickness *h*, as the gas-sensitive layer of FPR, is applied to the end face of the optical fiber [26]. The film thickness and the polymeric material are selected based on the type of gas under test. The thin film refractive index depends on the tested gas concentration. The sensors considered in this paper utilize a standard single-mode optical fiber with a core diameter of 8.2 μm, and the core and cladding refractive index of 1.4682 and 1.45, respectively.

The manufacturing of ACFOSs can be performed using two technologies similar to the CFOS fabrication technique. Using an ultraviolet continuous laser, the process starts with the two FBG forming. Then, FPR is formed at the fiber end face. In some cases, a high temperature is required for FPR forming, which can “erase” FBG or its part. For this reason, FPR is usually located at a distance from 1 to 4 cm from AFBSs. To reduce this distance, the FBG streaming technique can be used [27,28]. First, FPR is formed at the fiber end face, and then a femtosecond laser is focused through a glass capillary into the core of the fiber for AFBS forming.

To measure the greenhouse gas concentration, the FPR gas-sensitive layer is formed from different polymers: PEI/PVA—for CO_2_ [20]; polyaniline/Co_3_O_4_—for CO [29]; LuBc_2_—for NO_2_; PDMS/PMMA—for NH_3_ [30]; cryptophane A—for CH_4_; and cellulose—for O_x_. Humidity measurement can be performed using PVA coating [31]. It must be noted that the choice of polymers is not limited to the abovementioned types, and the sensors utilizing other polymers can be developed as well. The film thickness is tens of microns. AFBSs are formed at the 1550 nm range, due to both the advanced elemental base for the telecommunication systems and the spectral response of polymers and gases in this range.

## 3. The ACFOS Principle

ACFOSs consist of AFBSs combined with FPR, which is formed as a polymer film at the fiber end face, as shown in Figure 1. FPR consists of two reflective surfaces: the interface between the fiber core and the polymer film (interface 1), and the interface between the polymer film and environment (interface 2). The directed broadband light is initially reflected from AFBSs and then the traversing light is reflected from FPR. Two reflected beams interfere with each other due to the phase delay caused by the difference in optical paths. Since the reflectivity of the optical fiber end face and the surface of the polymer film are weak, the effect of multiple reflections can be neglected. This only allows to consider the first-order reflected beams. Consequently, the intensity of the ACFOS output light can be expressed as a combined spectrum, similarly to [32]:(1)$$I_{\mathrm{out}}≃\;I_{\mathrm{in}}\left[\rho_{1}+\rho_{2}+{\left(1-\rho_{1}-\rho_{2}\right)}^{2}\rho_{\mathrm{FP}}\right]$$
where ρ_1_, ρ_2_ and ρ_FP_ are the spectral reflectances of the FBG components of AFBSs and FPR, which are defined as [32]:(2)$$\rho_{i}=R_{i}\exp\left[-{\left(\lambda -\lambda_{i}\right)}^{2}/\omega^{2}\right]$$
and
(3)$$\rho_{\mathrm{FP}}=\;2R_{\mathrm{SMF}}\left[1+\cos\left(4\pi L/\lambda +\pi \right)\right]$$
where *R_i_* are the peak reflectances of the first and second FBG*_i_* (*i* = 1,2, *R_i_* = 0.6 ÷ 0.8), λ*_i_* are their central wavelengths and ω is bandwidth of the FBG, *R*_SMF_ is the reflectance of the optical fiber end face and *L* is the FPR length [33]:(4)$$L=\lambda_{1}\lambda_{2}/2\left(\lambda_{2}-\lambda_{1}\right)$$
where λ_1_, λ_2_ are the wavelengths of the neighboring maximums of the FPR reflectance spectrum comb.

It is possible to conclude that the resulting ACFOS spectrum is the superposition of the AFBS and FPR spectra. The sensor response to the gas concentration change can be related with the elastic-optical effect, which leads to the change in the FPR polymer film thickness. The temperature response of the sensor can be explained by the thermal expansion effect and the thermo-optic effect of the polymer. The thermo-optic effect also changes the FPR wavelength and simultaneously changes the AFBS component wavelengths. The central wavelength shift of the FBG components of AFBSs is defined as [22]:(5)$$\Delta \lambda_{i}=\;2\delta n_{\mathrm{eff}}\Lambda_{i}$$
where Λ*_i_* are periods of homogeneous sections of the first and second FBG, and δ*n*_eff_ is the change in the effective refractive index of the optical fiber core.

The simultaneous measurement of the gas concentration and the temperature can be made by measuring the comb wavelength of FPR and the central wavelength shift of the FBG components of AFBSs. When the wavelengths of the FPR spectral response Δλ_FP_ and the central wavelengths of the FBG components of AFBS Δλ_AFBS_ (Δλ_AFBS_ = Δλ_1_ = Δλ_2_, as it follows from the AFBS theory [22]) are determined, the matrix of the sensor sensitivity can be constructed:(6)$$\left[\begin{array}{c} \Delta \lambda_{\mathrm{FP}}\\ \Delta \lambda_{\mathrm{AFBS}} \end{array}\right]=\left[\begin{array}{cc} K_{\mathrm{FP},C} & K_{\mathrm{FP},T}\\ K_{\mathrm{AFBS},C} & K_{\mathrm{AFBS},T} \end{array}\right]\left[\begin{array}{c} \Delta C\\ \Delta T \end{array}\right]$$
where *K*_FP,C_ and *K*_FP,T_ are FPR sensitivities, and *K*_AFBS,C_ and *K*_AFBS,T_ are AFBS sensitivities for the gas concentration and the temperature, respectively. The relative values of the gas concentration and the temperature can be obtained from the sensitivity matrix:(7)$$\left[\begin{array}{c} \Delta C\\ \Delta T \end{array}\right]=\frac{1}{M}\left[\begin{array}{cc} K_{\mathrm{AFBS},T} & -K_{\mathrm{FP},T}\\ -K_{\mathrm{AFBS},C} & K_{\mathrm{FP},C} \end{array}\right]\left[\begin{array}{c} \Delta \lambda_{\mathrm{FP}}\\ \Delta \lambda_{\mathrm{AFBS}} \end{array}\right],$$
where
(8)$$M=K_{\mathrm{AFBS},T}K_{\mathrm{FP},C}-K_{\mathrm{AFBS},C}K_{\mathrm{FP},T},$$
is the determinant of the sensitivity matrix.

The matrix coefficients can be determined by measuring the sensor characteristics separately for the temperature and gas concentration by calibrating the wavelength comb of the FPR spectrum and the AFBS central wavelength.

Simulations of the spectral response of ACFOSs with 1 pm resolution between 1520 and 1580 nm showed that the spectrum (at room temperature and 1 atm pressure) shows ultrahigh spectral contrasts of approximately 18 and 9 dB for FPR and AFBSs, respectively. The reflected spectrum is shown in Figure 2 for the case of a typical concentration of carbon dioxide in the air.

## 4. ACFOS Modeling

As mentioned above, the AFBS consists of the two homogeneous FBGs with close but not equal central frequencies. AFBSs can be formed with two FBGs inscribed sequentially or on the same section of the optical fiber [34]. At the same time, it is required that the AFBS has spectral components, the full width at half maximum (FWHM) of which is much smaller than the address frequency of the AFBS [22]. The FWHM of the AFBS spectral components define the width of the address frequency spectrum at the photodetector’s output, which in turn defines the calculation accuracy of the AFBS central wavelength shift. Therefore, the bigger the AFBS physical length, the higher its accuracy is. AFBSs formed by two sequential FBGs require the very precise temperature control of the fiber section incorporating the AFBSs in order to ensure the constancy of the address frequency. This condition is not always easily met, especially when the AFBS length is several dozens of millimeters. Therefore, the inscription of the two homogeneous FBGs on the same fiber section is a more promising solution.

Despite the presence of the common assumptions and conclusions from the FBG theory, it is preferable to have a complete mathematical model which could simulate the ACFOS as a whole. The model of a plane electromagnetic wave propagation through a layered structure can be used as a basis for modeling of the optical radiation propagation through the ACFOS [35]. Indeed, an FBG can be represented as a layered structure consisting of numerous homogeneous transparent layers of silica glass with an alternate refractive index. The AFBS model is formed by two FBGs with different grating periods located sequentially or in the same fiber section. FPR is modelled as the final homogeneous transparent layer with the parameters of the gas-sensitive film. The Cartesian coordinate system with the origin at the beginning of the AFBS is chosen for modeling [35]. Then, for the plane electromagnetic wave propagation, the continuity of the electromagnetic field is required. This implies that the electric and magnetic fields are equal at each optical layer interface:(9)$$\left\{ \begin{array}{l} E_{i}\left(z_{i}\right)=E_{i+1}\left(z_{i}\right)\\ H_{i}\left(z_{i}\right)=H_{i+1}\left(z_{i}\right) \end{array}\right.,\;i=0,N-1,$$
where *E_i_* and *H_i_* are the vector magnitudes of the electric and magnetic fields, respectively; *i* is the layer number, so that *i* = 0 is the layer of silica fiber from the optical source to the AFBS, *i* = 1, *N* − 2 are the layers of alternate refractive index, *i* = *N* − 1 is the layer of the gas-sensitive film, and *i* = *N* is the layer of the environment. For the plane electromagnetic wave, the system of Equation (9) is formulated as follows:(10)$$\left\{ \begin{array}{l} t_{i}\cdot e^{-j\kappa_{i}z_{i}}+r_{i}\cdot e^{j\kappa_{i}z_{i}}=t_{i+1}\cdot e^{-j\kappa_{i+1}z_{i}}+r_{i+1}\cdot e^{j\kappa_{i+1}z_{i}}\\ \frac{t_{i}\cdot e^{-j\kappa_{i}z_{i}}-r_{i}\cdot e^{j\kappa_{i}z_{i}}}{w_{i}}=\frac{t_{i+1}\cdot e^{-j\kappa_{i+1}z_{i}}-r_{i+1}\cdot e^{j\kappa_{i+1}z_{i}}}{w_{i+1}} \end{array}\right.,\;i=0,N-1.$$

Here, *t_i_* and *r_i_* are the transmittance and reflectance of each layer, *z_i_* is the coordinates of the interfaces between layers, κ*_i_* is the wavenumber and *w_i_* is the wave impedance of each layer:(11)$$\begin{array}{l} \kappa_{i}\left(\lambda ,\varepsilon_{i},\mu_{i}\right)=\frac{2\pi }{\lambda }\sqrt{\varepsilon_{i}\mu_{i}}=\frac{2\pi }{\lambda }n_{i},\\ w_{i}\left(\mu_{i},\varepsilon_{i}\right)=\sqrt{\mu_{i}\mu_{0}/\varepsilon_{i}\varepsilon_{0}} \end{array}$$
taking into consideration that $c\sqrt{\varepsilon_{0}\mu_{0}}\equiv 1$, where λ = c/*f* is the wavelength, ω = 2π*f* is the circular frequency, ε*_i_* is the permittivity and μ*_i_* is the permeability of the layer, which are determined by the corresponding coefficients of the layers multiplied by the absolute permittivity ε_0_ and the absolute permeability μ_0_ of the vacuum, and c is the light speed in vacuum.

The system of Equation (10) provides 2∙*N* equations for finding the 2∙(*N* + 1) unknown quantities. It is assumed that all wave intensity that comes from the light source into the first layer passes through it without loss and there is no reflection from the far boundary of the third layer. These conditions make it possible to determine the reflection and transmission coefficients for the zero and the last layers (*t*_0_ = 1, *r_N_*_+1_ = 0). Hence, we obtain a system of 2∙*N* linear equations for 2∙*N* unknown complex variables *r_i_*, *t_i_*.

The solution of the system of equations allows to define the transmittance and reflectance for each layer at any wavelength. The solution of the system of equations for each wavelength in a particular range enables the complete modeling of the AFBS reflectance spectrum.

The system of Equation (10) was initially formulated on the condition that the layer interfaces coincide with the chosen coordinate grid *z_i_*. In this case, as it was mentioned before, *z_i_* is chosen so that at *i* = 1, *N* – 2, they would form layers with alternate refractive indices. Such a choice of the coordinate grid is convenient for the homogeneous layer calculation. However, this approach is problematic in the case of modeling two periodic structures formed on top of each other, since it is not possible to impose two periodic structures with different periods on the same coordinate grid.

On the other hand, the conditions of electromagnetic field continuity are not only met at the interface between layers but also in every point of electromagnetic wave propagation. This fact allows to use the coordinate grid *z_i_* that does not coincide with the layers’ interfaces.

The induced refractive index of the optical fiber changes harmonically. Therefore, if the coordinate grid does not coincide with the layers of periodic structure, it is necessary to define the value of the induced refractive index for each layer. For that, it is sufficient to define the permittivity of each layer, superimposing the periodical variation of *ε_i_* on the arbitrary uniform coordinate grid under the assumption that the permeability of all the layers is equal to one:(12)$$\varepsilon_{i}=\left(\left(n_{0}^{2}+n_{1}^{2}\right)+\frac{\pi }{2}\left(n_{0}^{2}-n_{1}^{2}\right)sin\left(\frac{2\pi }{\Lambda }\cdot \frac{z_{i+1}+z_{i}}{2}\right)\right)\frac{1-j\cdot tan\left(\alpha \right)}{2}$$
where *n*_0_ and *n*_1_ are the refractive indices of the FBG layers, Λ is the FBG period, tan(α) is the dielectric loss tangent of the optical fiber and *z_i_* is the arbitrary coordinate grid.

The convenience of using the arbitrary coordinate grid lies in the fact that by adding other summands with different periods of harmonic variation of the refractive index to Equation (12), it is possible to model any number of FBGs inscribed on the same section of the optical fiber. The use of the coordinate grid which is not linked up to the FBG period makes it possible to model various ACFOS configurations with different address frequencies and to obtain the values of the induced refractive index change along the optical fiber for ACFOS point-by-point inscription with the desired spectral response.

Figure 3 shows the results of the numerical simulation of the ACFOS spectral response using the mathematical model introduced above. The address frequency of ACFOS is 80 GHz.

As it can be seen from Figure 3, the ACFOS spectrum response is comprised of a sinusoid-like spectral shape formed by the Fabry–Perot cavity and two narrow peaks formed by FBGs. It must be noted that the spectral response contains dips near the reflection peaks of FBGs, which can be explained by the interference of waves reflected from the FBGs, which received a phase shift when passing through the Fabry–Perot cavity.

Using the proposed mathematical model, it is possible to study the nonuniform heating of the ACFOS along its length to analyze the usage of chirped and complex apodized FBGs in the structure of ACFOSs. The model also enables studying the spectral response variation due to the gas concentration and temperature influence on the sensor.

## 5. ACFOS Multiplexing

Sensor multiplexing is based on the address measuring conversion for the AFBS, namely: “the AFBS central wavelength—the difference frequency between the first and the second FBG components of AFBS—the photodetector output beat frequency—the AFBS microwave frequency address”. Simulations of the AFBS sensors with different address frequencies were performed. The dependence of the address frequencies on the grating period of the constituent FBGs in the range up to 20 GHz was obtained (Figure 4). The address frequencies of 8, 15.6 and 19.2 GHz were obtained. The dependence of AFBS spectral response on the FBG length was also established, as shown in Figure 4d. With the increase in FBGs lengths, their reflectance also increases. Therefore, a tradeoff between the sensor length and the reflectance must be achieved. The dependence clearly demonstrates the possibility of sensor multiplexing in a narrow wavelength range compared to the operating wavelength of 1550 nm with full addressability.

Numerical experiments for the multisensory gas concentration control system were performed in the Optiwave System software. The sensor addressability was determined by the nonequivalent conditions of the AFBS address frequencies Ω*_k_* ≠ Ω*_j_*, where *k* and *j* are the indexes of the AFBS in the sensor array, *k*, *j* ∈ *N* and *N* is the number of AFBSs. The difference |Ω*_k_* − Ω*_j_*| must also not be equal or a multiple of either of the address wavelengths Ω*_k_* and Ω*_j_* [24].

The frequency difference between the two neighboring sensors is approximately 100 MHz. If the address frequencies of ACFOSs do not exceed 20 GHz and the sensor number is no more than 200, then they can be differenced by the fast Fourier transform. The sensor number will be less if the bandwidth is decreased to 120 MHz (which corresponds to the FBG recording technology with a minimum bandwidth of 1 pm [36]). The existing methods of determining the AFBS central wavelength based on the optical filter with an inclined profile [22] and based on the fast Fourier transform [25] allow determining the absolute temperature with an error not exceeding ±0.1 °C and ±0.01 °C, respectively. The main condition for accurate temperature determination is the equality of the reflection coefficients *R_i_* of both FBGs forming ACFOSs. The ACFOS spectrum is a superposition of the AFBS and FPR spectra, and hence, the condition of equality is not fulfilled.

The Karhunen–Loève transform (KLT) can be used to solve this problem [37]. The ability of KLT to separate the FBG and the FPR spectra can be successfully used for the interrogation of the ACFOS spectrum. This is a common procedure used in multisensory systems for oil, gas and geothermal engineering, where the combined sensors operate in the temperature range from −60 to 300 °C and at high pressure (up to 100 atm). The KLT procedure is also applicable for environmental monitoring, where the conditions are similar in temperature and in gas concentration changes from 10 to 90%.

The data separation from seven sensors is demonstrated in [21]. The first five sensors are the FBGs with similar reflection coefficients. They are uniformly distributed in the 1550 nm range, and have a different reflection amplitude at the central wavelength due to the overlap with the FPR spectrum. The sixth sensor is the external FPR with a maximum reflectance of 33% and the resonator length of 25 μm is designed to control the external influences. Finally, the seventh sensor is the weakly reflective FPR (0.95%) which simulates a pressure sensor. The total ACFOS reflectance spectrum has a width of 60 nm. After KLT is applied, the components of each sensor are separated, while all FBGs have equal reflection amplitudes.

The KLT algorithm use can change the structure of wave multiplexing multisensory systems, particularly for high-density sensors [38]. The typical principle of such systems is based on the wavelength separation, assigned as the sensor work range. This approach is very vulnerable due to the spectra overlapping possibility. We eliminate this disadvantage by using AFBSs, which can also operate at the common central wavelength [24]. In addition, the use of the KLT algorithm allows eliminating the ambiguity of the sensor readings, since each sensor is encoded in its spectrum part.

The discussion in this section represents the first step towards combining the address-based approach in ACFOSs with the KLT algorithm. The goal statement for further research is to investigate the possibility of applying of two-, three- and four-component AFBSs [24,25,38] with a strongly reflective FPR for the temperature-compensated gas concentration measurements and with a weakly reflective FPR for ambient pressure compensation.

## 6. Multisensory Environmental Monitoring System

Let us focus on the task statement of the ACFOS multisensory system creation, the core of which must be an interrogator. The design of the interrogator is important, not only because it determines the system performance and the measuring conversion principle, but also because it must ensure its working ability not only in the laboratory, but also in the field conditions. For ACFOSs, the system requirements are as follows.

*Physical requirements:* The optical fibers guarantee small sensor size, easy-to-lay cable and lightweight construction. Small size is essential for environmental applications where the key requirement is minimizing the control point size.

*Metrological requirements:* Most ACFOSs provide a high conversion accuracy, linear calibration function and fast response.

*System requirements:* ACFOSs enable multisensory architectures in which sensors can be fabricated on a single fiber or combined across different topologies into a single interrogation system. ACFOS allows constructing the quasi-distributed measurement systems with high spatial resolution.

*The field measuring systems must comply with the following requirements:* The interrogator cost must be limited by its specific application: the size, weight and shape of the interrogator must match the size of the rack of the automatic air pollution monitoring station; it must have low-power consumption with the battery power backup; the data recording rate for each channel must be 10 times more than the possible variation of the measured value; and the number of ACFOSs must be determined by the measurement requirements.

The most important factor of any measuring system is its cost. The cost depends on many factors. The subsystem that has a dominant part of the cost is usually determined by its functional purpose. In some applications, a fiber cost may be dominant, whilst in others it may be the multiplexing scheme. In sensor systems, the interrogator typically has a dominant part of the cost. In [22,25], we showed that the cost of the microwave photonic interrogators for AFBSs can be in tens of times less than the classical optical-electronic interrogator cost. Consequently, the main challenge for the system developers is to obtain information about the FPR shift on the address frequencies of the sensor using the KLT algorithm.

The structural diagram of the designed system working on the reflection principle is shown in Figure 5. Sensors (6.1)–(6.N) are mounted remotely, and the microwave photonic interrogator is mounted in the automated control rack. The light source (1) is a broadband laser diode (LD) in the range of 1550 nm. Its temperature must be stabilized by the thermoelectric controller, which is critical for ensuring the stable output power of the light source over the operating wavelength range. The optical radiation from the source (1) is divided into two channels by means of an fiber-optic splitter (2) in order to compensate for possible instabilities in the LD output power through the normalization of the signal amplitudes in both channels. In channel 1, the reflected radiation from ACFOSs (6.1)–(6.N) connected through splitter (5) is directed to the photodetector (7.1) through the optical circulator (4.1), while source (1) is isolated from the reflected radiation by optical isolator (3.1) (>20 dB). Each ACFOS can use FPR with different polymer films. The output signal of the photodetector (7.1) is converted by means of the analog-to-digital converter (ADC) (8.1) and processed by the computer (10). Similarly, in channel 2, the radiation reflected from the reference sensor (9) is guided to the photodetector (7.2) through the circulator (4.2), while the source is isolated from it with the isolator (3.2). The reference sensor (9) has the reference spectrum equal to the spectrum of the measurement sensor at the calibration points. This allows the additional consideration of variations in LD parameters. The output signal of the photodetector (7.2) is also converted by the second ADC (8.2) and processed by the computer (10).

The ACFOS packaged design in the ferrule is shown in Figure 6a. The overall design of the microwave photonic interrogator (MPI), considering the functional electronics and the commutation fibers, is shown in Figure 6b. In addition, the Ibsen I-MON optical-electronic interrogator, used to control and compare the measurement results, is installed in housing.

MPI consists of the following nodes:(1)Optical-electronic module (OM) of microwave photonics type;(2)Cross-1 (KP1) (for switching the optical-electronic module with sensors through a fiber-optic cable);(3)Cross-2 (KR2) (for splitting the fiber-optic cable for 24 channels for sensor connecting);(4)Fiber-optic cable connecting the KP1 and KP2;(5)Patch cords for connecting the RC2 and ACFOS.

The use of non-electrical measuring instruments and the fiber-optic cable allows applying MPI for environmental monitoring in the energy, oil and gas as well as chemical industries (including manufacturing with aggressive gases), metallurgical enterprises and in medicine.

ACFOSs that are not included in the MPI kit consist of electrically non-conductive materials, which enables them to be used in high-voltage areas. In addition, they are manufactured in the enclosure, which is not susceptible to corrosion. ACFOSs are also immune to electromagnetic noises and do not interact with other electrical devices. They can be safely used in potentially explosive environments without the risk of sparks. The sensors have a high accuracy and wide operating temperature range from −60 to +300 °C, depending on the type of fiberglass cable protection. If necessary, it is possible to combine (multiplex) numerous sensors into one measurement network, with the measurement module, placed at the distance of up to 10–30 km. Estimations have shown that the ACFOS design and system as a whole allows measuring the gas concentration in the range of 10–90% with an error of 0.1–0.5%.

The undoubted advantages of ACFOSs are their high speed, passivity, resistance to electromagnetic noises, dielectric nature, fire safety, low weight and dimensions, operability in a wide temperature range and interference immunity of the data transmission channel. This approves the possibility of building on their basis, promising tools and multisensor systems for the environmental monitoring of green-house gas concentrations.

## 7. Conclusions

As the result of this research, the concept of using multisensory systems for the environmental monitoring of greenhouse gases at various levels was developed. The concept allows forming a deployed network of control posts for enterprises, regional formations and the whole country by 2030. The proposed concept is based on the current level of information technology and computational tools, which enables the improvement of systems for monitoring and managing of the environmental situation. Intelligent technologies and monitoring means based on fiber-optic technologies should become the priority directions for increasing the ecological safety level. To create an intelligent system for greenhouse gas monitoring, it is first necessary to solve the following tasks under field conditions:—The address CFOSs, based on AFBSs and FPR, capable of distinguishing the gas type and determine its concentration, have a built-in system of temperature, atmospheric pressure and ambient humidity compensation;—The microwave photonic interrogator for the construction of multisensory system of ecological greenhouse gas monitoring possesses the possibility of controlling the ACFOSs address and analysis reflection from them in the narrowband, is limited by AFBS address frequencies, with is transformed into a digital data package;—The multisensory system software to process a digital data package allows the separate registering of AFBS and FPR responses for gas concentration, temperature, pressure and ambient humidity based on the single multi-parameter sensitivity matrix and Karhunen–Loève transformation algorithms to eliminate cross-distortions;—The passive fiber-optic communication network of a hybrid structure with time and wave multiplexing, providing data exchange channels between an ACFOS and dispatch center, with the possible use of wireless access networks;—The artificial intelligence technology processes of large amounts of data for operational decision making to ensure the maximum level of environmental safety at various levels of the system.

Each of the aforementioned tasks requires the scrupulous study and presentation of the detailed results obtained at each stage of implementation, which is the subject of our further research. In this article, the authors did not go into detail about the measurements performed, but focused on the problem statements, excluding the last two. Expanded information will be the subject of future publications.

## Figures and Tables

**Figure 1 sensors-22-04827-f001:**
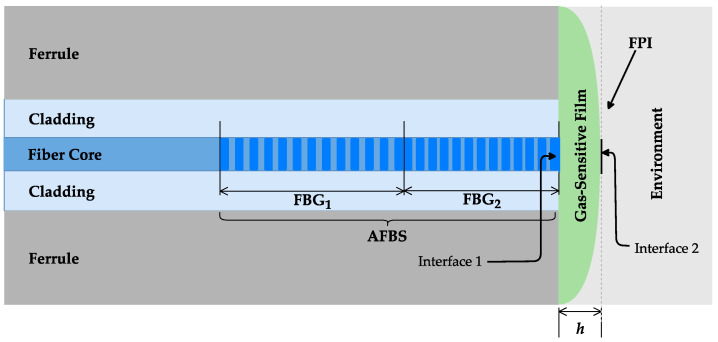
The structural diagram of an ACFOS.

**Figure 2 sensors-22-04827-f002:**
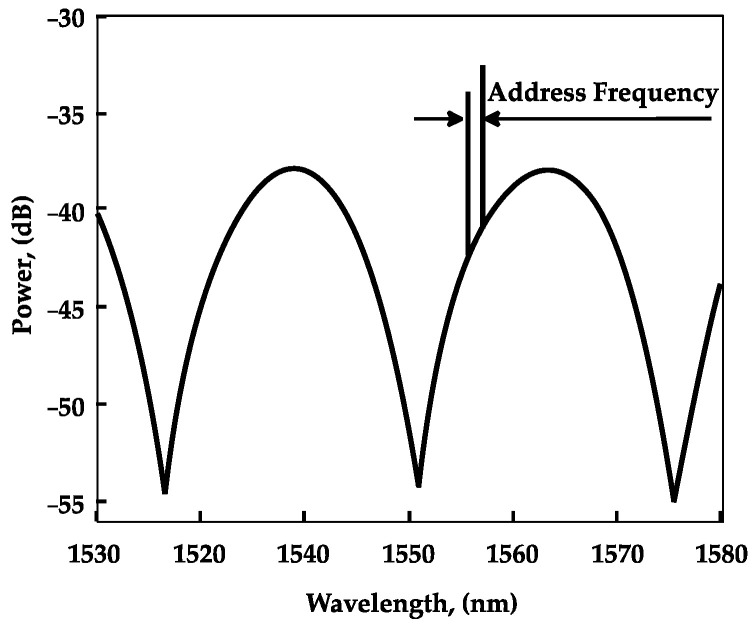
CFOS reflected radiation spectrum (laboratory conditions).

**Figure 3 sensors-22-04827-f003:**
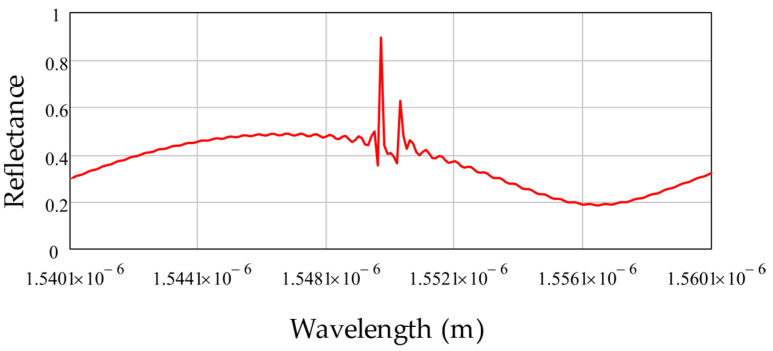
ACFOS reflected radiation spectrum (numerical simulation).

**Figure 4 sensors-22-04827-f004:**
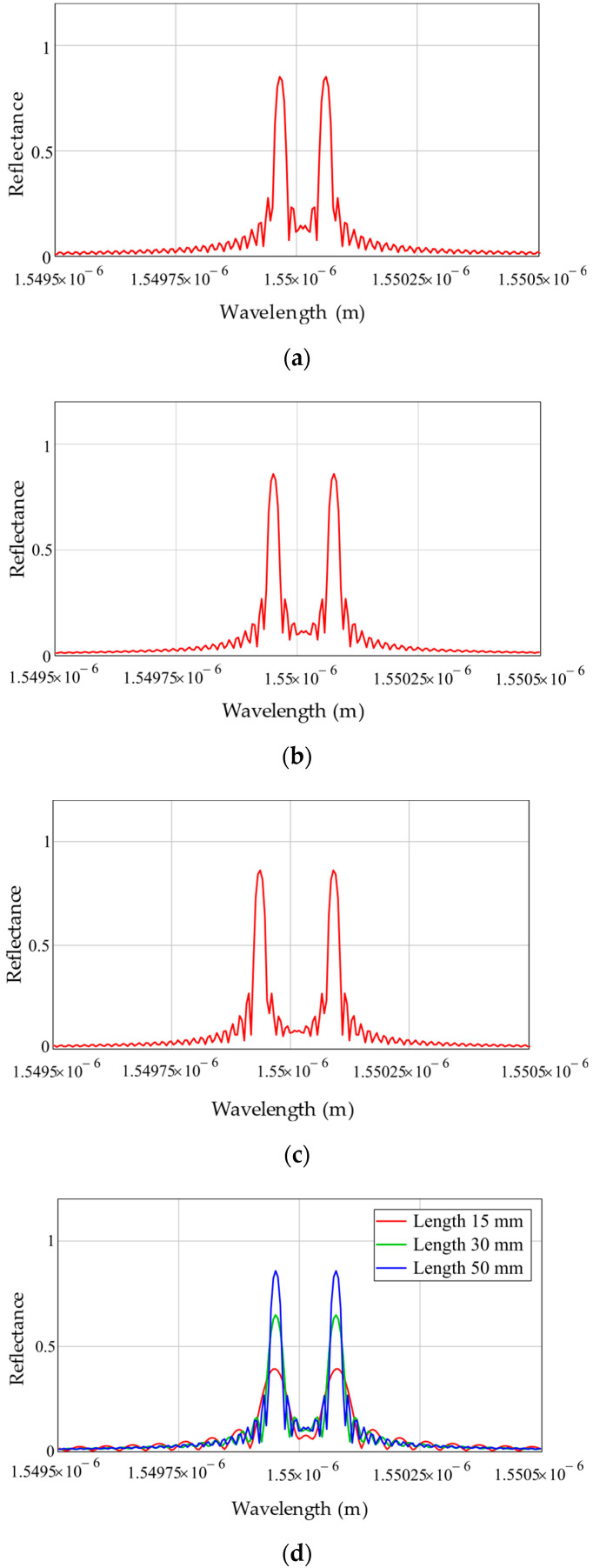
Simulation results of AFBS with two FBGs. AFBS address frequencies: (**a**) 8 GHz; (**b**) 15.6 GHz; (**c**) 19.2 GHz; and (**d**) length variation of both FBGs.

**Figure 5 sensors-22-04827-f005:**
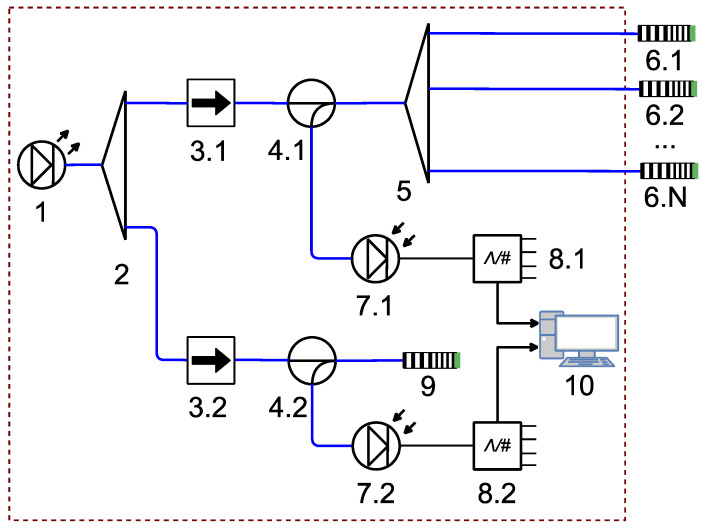
Schematic diagram of a multisensor greenhouse gas monitoring system: (1) wideband light source; (2) and (5) fiber-optic splitters; (3.1) and (3.2) optical isolators; (4.1) and (4.2) fiber-optic circulators; (6.1)—(6.N) ACFOSs; (7.1) and (7.2) photodetectors; (8.1) and (8.2) ADCs; (9) reference sensor; and (10) computer; blue lines indicate optical connections; black lines indicate electrical connections.

**Figure 6 sensors-22-04827-f006:**
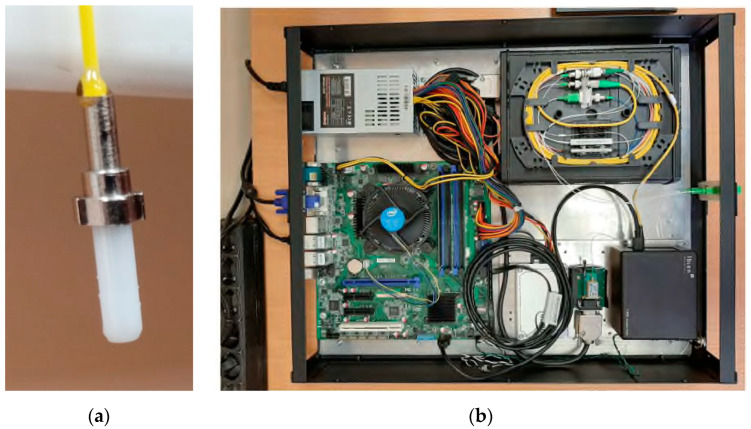
A prototype of the CFOS (**a**) and interrogator (**b**) for a multisensor system.

## Data Availability

The data presented in this study are available on request from the corresponding author. The data are not publicly available due to rules of our contract conditions with our customer.

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
