# Peer review of "Addressed Combined Fiber-Optic Sensors as Key Element of Multisensor Greenhouse Gas Monitoring Systems"

_sensors, 2022, doi:10.3390/s22134827_

Round 1
Reviewer 1 Report
The comments on the manuscript entitled: "Addressed Combined Fiber-Optic Sensors as Key Element of Multi-Sensor Greenhouse Gas Monitoring Systems" by Morozov et al.:
1. The manuscript needs to be carefully edited. For example in figure caption 6: "Figure 6. Laboratory layout of CFOS (a) and interrogator (b) for a multi-sensor system". And in Line 16: " ... the combined fiber-optic sensors (CFOSs), ..."
2. The manuscript focuses on the combined fiber optic sensors, therefore it is recommended to address some important parameters of these sensors, such as detection limit, quality factor, and sensitivity, in addition to the environment temperature.
3. The physical parameters of the used fiber (such as core diameter and the core and cladding refractive index) should be added to more clear Figure 1.
4. Please give proper references for Eq. (2)-(4), and (10).
5. Is there any experimental results for numerous sensors into one measurement network placed at the distance up to 10−30 km?
6. Some results need to be analyzed, e.g. Figure 3.
7. The combined fiber-optic sensors consist of the fiber Bragg grating and the Fabry-Perot resonator. Instead of FBG, the recent photonic crystals can be used with better performance. It is suggested to describe and to address this mater.
Author Response
First of all, we wish to thank the anonymous Reviewer for the precious comments that allowed us to improve the quality of the paper.
- The manuscript needs to be carefully edited. For example in figure caption 6: "Figure 6. Laboratory layout of CFOS (a) and interrogator (b) for a multi-sensor system". And in Line 16: " ... the combined fiber-optic sensors (CFOSs), ..."
Thank you for the remark. The mentioned expressions were corrected.
- The manuscript focuses on the combined fiber optic sensors, therefore it is recommended to address some important parameters of these sensors, such as detection limit, quality factor, and sensitivity, in addition to the environment temperature.
The sensors proposed in this manuscript were tested at the environmental temperature of 22°C and demonstrated the measurement of CO2 gas concentration in the range of 10…90% with the accuracy of 0.1–0.5%. The sensitivity of the sensors is ~0.25 nm/% and is similar to the one of the sensors that also utilized PEI/PVA coating in the reference [29]. The FWHM of the spectral components of the addressed fiber Bragg structures is ~30 pm.
- The physical parameters of the used fiber (such as core diameter and the core and cladding refractive index) should be added to more clear Figure 1.
Thank you for the suggestion. The following sentence was added for clarification (lines 144-146): “The sensors considered in this paper utilize a standard single-mode optical fiber with the core diameter of 8.2 μm, and the core and cladding refractive index of 1.4682 and 1.45, respectively.”
- Please give proper references for Eq. (2)-(4), and (10).
We thank the Reviewer for the remark. The citations for equations (2)-(4) were added to the text. The equation (10) was formulated by the authors as it follows from the equation (9) due to the conditions of the continuity of an electromagnetic wave propagation that are ensured by the equality of the electric and magnetic fields at the interfaces between the media.
- Is there any experimental results for numerous sensors into one measurement network placed at the distance up to 10−30 km?
At this point, we propose a new concept of combined fiber-optic sensors, which possess the intrinsic advantages of fiber-optic sensors, such as the possibility to merge numerous sensors into a single network as well as to perform long-distance measurement. However, the experiments on the remote measurements using the multi-sensory systems will be in the scope of further research.
- Some results need to be analyzed, e.g. Figure 3.
The following comments were added regarding the Figure 3 (lines 294-298): “As it can be seen from Figure 3, the ACFOS spectrum response is comprised of a sinusoid-like spectral shape formed by the Fabry-Perot cavity and two narrow peaks formed by the FBGs. It must be noted that the spectral response contains dips near the reflection peaks of the FBGs, which can be explained by the interference of waves reflected from the FBG, which received a phase shift when passing through the Fabry-Perot cavity.”
- The combined fiber-optic sensors consist of the fiber Bragg grating and the Fabry-Perot resonator. Instead of FBG, the recent photonic crystals can be used with better performance. It is suggested to describe and to address this mater.
The current paper focuses on the application of fiber Bragg gratings in the combined sensors with Fabry-Perot resonators. Although the usage of photonic crystals in such sensors is also possible, it requires a separate investigation and will be covered in further works.
Reviewer 2 Report
Dear authors
This research used Fiber-optic sensors for design of multi-sensor greenhouse gas monitoring system. I think that this concept is of great interest to the community. However, it should be significantly upgraded before publishing on the Journal.
1. This is a reseach article, but its structure is not appropriate. The authors should re-construct it, following 4 main contents: Introduction, Methodology, Results and discussions, Conclusions
2. This is a multi-gas sensors. It is difficult to correctly determine the sensitivity and selectivity of this sensor system, which is very important for a gas sensor in practical applications.
3. The authors should conduct a selective test with different gases like NH3 and VOCs
4. Although Introduction is long, it not emphasize the importance of the approach in this research. Furthermore, a litterature review about the similar approaches or multi-gas sensor systems should be summarized and compared
Author Response
We would like to thank the Reviewer for the comprehensive review of the paper and the valuable remarks.
- This is a reseach article, but its structure is not appropriate. The authors should re-construct it, following 4 main contents: Introduction, Methodology, Results and discussions, Conclusions
The authors believe that the chosen naming of the article sections reflects their contents, since different sections are dedicated to different aspects of the sensors and sensor systems. Methodology is in the scope of the Section 2 ACFOS Model and Section 3 The ACFOS Principle. Results and discussions are covered in the Section 4 ACFOS Modeling, Section 5 ACFOS Multiplexing, and Section 6 Multisensory Environmental Monitoring System.
- This is a multi-gas sensors. It is difficult to correctly determine the sensitivity and selectivity of this sensor system, which is very important for a gas sensor in practical applications.
The test results of the sensors proposed in this manuscript demonstrated the measurement of CO2 gas concentration in the range of 10…90% with the accuracy of 0.1–0.5%. The sensitivity of the sensors is ~0.25 nm/% and is similar to the one of the sensors that also utilized PEI/PVA coating in the reference [29]. The multi-gas sensing can be provided by using multiple sensors with different polymeric coatings.
- The authors should conduct a selective test with different gases like NH3 and VOCs
The tests conducted in the current research were dedicated to the measurement of CO2 concentration. The sensors can be applied for concentration measurement of other gases as well, however, the other types of polymer coating must be used. The tests with different gases like NH3 and VOCs will be conducted in further works.
- Although Introduction is long, it not emphasize the importance of the approach in this research. Furthermore, a litterature review about the similar approaches or multi-gas sensor systems should be summarized and compared
Thank you for the remark. The following sentence was added to emphasize the aim of this work (lines 102-104): “In order to eliminate the abovementioned problems while preserving the functional advantages, the current work proposes the usage of the Addressed Fiber Bragg Structures (AFBSs) instead of conventional FBGs.” The literature reviews of similar approaches were reported previously in numerous works, for example in [29-32, 38], therefore, the authors considered similar review to be redundant in the current paper.
Reviewer 3 Report
This paper presents a multisensory measurement system based on addressed fiber Bragg structures. It was not clear how the proposed, addressed fiber Bragg structures (AFBS) method measures various greenhouse gases without any cross-sensitivity. How each FBG monitor targeted gas, and discriminate against other greenhouse gases, even though the authors used one sensing film of polymeric material. Authors need to demonstrate the proposed AFBS scheme experimentally with varying gas concentrations in the presence of other gases. In the abstract, the authors stated, that measuring gas concentration in the range of 10…90% with the accuracy of 0.1–0.5%. Without any experiments, how do you conclude that the proposed multisensory FBG can measure within that range and accuracy? Although, the authors never stated which gas concentration can be measured in the range of 10…90%. Due to the lack of novelty, experimental analysis, and mismatch between abstract and experimental investigations, I suggest doing more emphasis on the proposed novelty with experimental investigations of various concentrations with their sensitivity, selectivity, readability, and response/recovery time using a multi-sensor system.
Author Response
We would like to thank the anonymous Reviewer for the comprehensive review of the paper and the valuable remarks.
This paper presents a multisensory measurement system based on addressed fiber Bragg structures. It was not clear how the proposed, addressed fiber Bragg structures (AFBS) method measures various greenhouse gases without any cross-sensitivity. How each FBG monitor targeted gas, and discriminate against other greenhouse gases, even though the authors used one sensing film of polymeric material. Authors need to demonstrate the proposed AFBS scheme experimentally with varying gas concentrations in the presence of other gases.
The elimination of cross-sensitivity is not in the scope of this paper. The aim of the manuscript is to propose the usage of addressed fiber Bragg structures instead of conventional FBGs in a well-established type of gas concentration sensors consisting of FBGs and a Fabry-Perot resonator that were reported in the reference [38], for example. The measurement of various gases can be conducted using different polymeric materials in the sensing film.
In the abstract, the authors stated, that measuring gas concentration in the range of 10…90% with the accuracy of 0.1–0.5%. Without any experiments, how do you conclude that the proposed multisensory FBG can measure within that range and accuracy? Although, the authors never stated which gas concentration can be measured in the range of 10…90%.
The reported gas concentration range and accuracy are based on the test results of CO2 concentration measurements, the results are similar to the ones of the sensors that also utilized PEI/PVA coating in the reference [29].
Due to the lack of novelty, experimental analysis, and mismatch between abstract and experimental investigations, I suggest doing more emphasis on the proposed novelty with experimental investigations of various concentrations with their sensitivity, selectivity, readability, and response/recovery time using a multi-sensor system.
The current manuscript proposes a new concept of combined fiber-optic sensors with Fabry-Perot resonators that utilize addressed fiber Bragg structures instead of conventional FBGs, which is stated in the Abstract as well as in the text of the paper. The test results of the sensors proposed in this manuscript were obtained for the measurement of CO2 gas concentration. The sensitivity of the sensors is ~0.25 nm/% and is similar to the one of the sensors that also utilized PEI/PVA coating in the reference [29]. The tests of the multi-sensor systems with different polymeric materials in the sensing film will be in the scope of further works.
Round 2
Reviewer 1 Report
According to my previous comments, the revised manuscript can be accepted for publication in Sensors.
Reviewer 2 Report
The manuscript can be accepted for publication in the current version
Reviewer 3 Report
The authors addressed the concerns and their responses are satisfactory. I still suggest formatting, for instance, a page. 10 alone have only large size figures with poor resolution. Use proper figure size with an acceptable resolution for all figures.